# Diffusion Based Representation Learning

## Abstract

Diffusion-based methods, represented as stochastic differential equations on a continuous-time domain, have recently proven successful as non-adversarial generative models. Training such models relies on denoising score matching, which can be seen as multi-scale denoising autoencoders. Here, we augment the denoising score matching framework to enable representation learning without any supervised signal. GANs and VAEs learn representations by directly transforming latent codes to data samples. In contrast, the introduced diffusion-based representation learning relies on a new formulation of the denoising score matching objective and thus encodes the information needed for denoising. We illustrate how this difference allows for manual control of the level of details encoded in the representation. Using the same approach, we propose to learn an infinite-dimensional latent code that achieves improvements of state-of-the-art models on semi-supervised image classification. We also compare the quality of learned representations of diffusion score matching with other methods like autoencoder and contrastively trained systems through their performances on downstream tasks.

## 1 Introduction

Diffusion-based models have recently proven successful for generating images (Sohl-Dickstein et al., 2015; Song & Ermon, 2020; Song et al., 2020), graphs (Niu et al., 2020), shapes (Cai et al., 2020), and audio (Chen et al., 2020b; Kong et al., 2021). Two promising approaches apply step-wise perturbations to samples of the data distribution until the perturbed distribution matches a known prior (Song & Ermon, 2019; Ho et al., 2020). A model is trained to estimate the reverse process, which transforms samples of the prior to samples of the data distribution (Saremi et al., 2018). Diffusion models were further refined (Nichol & Dhariwal, 2021; Luhman & Luhman, 2021) and even achieved better image sample quality than GANs (Dhariwal & Nichol, 2021; Ho et al., 2021; Mehrjou et al., 2017). Further, Song et al. showed that these frameworks are discrete versions of continuous-time perturbations modeled by stochastic differential equations and proposed a diffusion-based generative modeling framework on continuous time. Unlike generative models such as GANs and various forms of autoencoders, the original form of diffusion models does not come with a fixed architectural module that captures the representation.

Learning desirable representations has been an integral component of generative models such as GANs and VAEs (Bengio et al., 2013; Radford et al., 2016; Chen et al., 2016; van den Oord et al., 2017; Donahue & Simonyan, 2019; Chen et al., 2020a; Schölkopf et al., 2021). Recent works on visual representation learning achieve impressive performance on the downstream task of classification by applying contrastive learning (Chen et al., 2020c; Grill et al., 2020; Chen & He, 2020; Caron et al., 2021). However, contrastive learning requires additional supervision of augmentations that preserve the content of the data, and hence these approaches are not directly comparable to representations learned through generative systems like Variational Autoencoders (Kingma & Welling, 2013; Rezende et al., 2014) and the current work which are considered *fully* unsupervised. Moreover, training the encoder to output similar representation for different views of the same image removes information about the applied augmentations, thus the performance benefits are limited to downstream tasks that do not depend on the augmentation, which has to be known beforehand. Hence our proposed algorithm does not restrict the learned representations to specific downstream tasks and solves a more general problem instead. We provide a summary of contrastive learning approaches in Appendix A. Similar to our approach, Denoising Autoencoders (DAE) (Vincent et al., 2008) can be used to encode representations that can be manually controlled by adjusting the noise scale (Geras & Sutton, 2015;

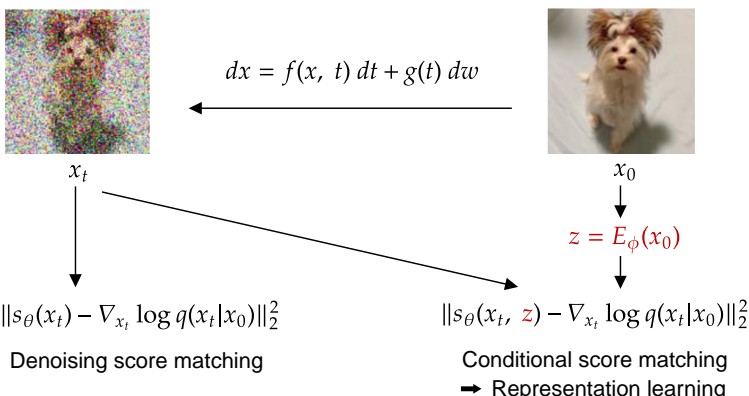

$$dx = f(x, t)\, dt + g(t)\, dw$$

$x_t$

$x_0$

$z = E_\phi(x_0)$

$\|s_\theta(x_t) - \nabla_{x_t} \log q(x_t|x_0)\|_2^2$

Denoising score matching

$\|s_\theta(x_t,\ z) - \nabla_{x_t} \log q(x_t|x_0)\|_2^2$

Conditional score matching
➡ Representation learning

Figure 1: Conditional score matching with a parametrized latent code is representation learning. Denoising score matching estimates the score at each $x_t$; we add a latent representation $z$ of the clean data $x_0$ as additional input to the score estimator.

Chandra & Sharma, 2014; Zhang & Zhang, 2018). Note that, unlike DAEs, the encoder in our approach does not receive noisy data as input, but instead extracts features based on the clean images. For example, this key difference allows DRL to be used to limit the encoding to fine-grained features when focusing on low noise levels, which is not possible with DAEs.

The main contributions of this work are

- We present an alternative formulation of the denoising score matching objective, showing that the objective cannot be reduced to zero.

- We introduce Diffusion-based Representation Learning (DRL), a novel framework for representation learning in diffusion-based generative models. We show how this framework allows for manual control of the level of details encoded in the representation through an infinite-dimensional code. We evaluate the proposed approach on downstream tasks using the learned representations directly as well as using it as a pre-training step for semi-supervised image classification, thereby improving state-of-the-art approaches for the latter.

- We evaluate the effect of the initial noise scale and achieve significant improvements in sampling speed, which is a bottleneck in diffusion-based generative models compared with GANs and VAEs, without sacrificing image quality.

## 1.1 Diffusion-based generative modeling

We first give a brief overview of the technical background for the framework of the diffusion-based generative model as described in Song et al. (2021b). The forward diffusion process of the data is modeled as an SDE on a continuous-time domain $t \in [0, T]$. Let $x_0 \in \mathbb{R}^d$ denote a sample from the data distribution $x_0 \sim p_0$, where $d$ is the data dimension. The trajectory $(x_t)_{t \in [0,T]}$ of data samples is a function of time determined by the diffusion process. The SDE is chosen such that the distribution $p_{0T}(x_T|x_0)$ for any sample $x_0 \sim p_0$ can be approximated by a known prior distribution. Notice that the subscript $0T$ of $p_{0T}$ refers to the conditional distribution of the diffused data at time $T$ given the data at time 0. For simplicity we limit the remainder of this paper to the so-called Variance Exploding SDE (Song et al., 2021b), that is,

$$\mathrm{d}x = f(x,t)\,\mathrm{d}t + g(t)\,\mathrm{d}w := \sqrt{\frac{\mathrm{d}[\sigma^2(t)]}{\mathrm{d}t}}\,\mathrm{d}w, \tag{1}$$

where w is the standard Wiener process. The perturbation kernel of this diffusion process has a closed-form solution being $p_{0t}(x_t|x_0) = \mathcal{N}(x_t; x_0, [\sigma^2(t) - \sigma^2(0)]I)$. It was shown by Anderson (1982) that the reverse diffusion process is the solution to the following SDE:

$$\mathrm{d}x = [f(x,t) - g^2(t)\nabla_x \log p_t(x)]\,\mathrm{d}t + g(t)\,\mathrm{d}\overline{w}, \tag{2}$$

where $\overline{w}$ is the standard Wiener process when the time moves backwards. Thus, given the score function $\nabla_x \log p_t(x)$ for all $t \in [0, T]$, we can generate samples from the data distribution $p_0(x)$. In order to learn the score function, the simplest objective is Explicit Score Matching (ESM) (Hyvärinen & Dayan, 2005), that is,

$$\mathbf{E}_{x_t} \left[ \|s_\theta(x_t, t) - \nabla_{x_t} \log p_t(x_t)\|_2^2 \right]. \tag{3}$$

Since the ground-truth score function $\nabla_{x_t} \log p_t(x_t)$ is generally not known, one can apply denoising score matching (DSM) (Vincent, 2011), which is defined as the following:

$$J_t^{DSM}(\theta) = \mathbf{E}_{x_0} \{ \mathbf{E}_{x_t|x_0} [\|s_\theta(x_t, t) - \nabla_{x_t} \log p_{0t}(x_t|x_0)\|_2^2] \}. \tag{4}$$

The training objective over all $t$ is augmented by Song et al. (2021b) with a time-dependent positive weighting function $\lambda(t)$, that is, $J^{DSM}(\theta) = \mathbf{E}_t \left[ \lambda(t) J_t^{DSM}(\theta) \right]$. One can also achieve class-conditional generation in diffusion-based models by training an additional time-dependent classifier $p_t(y|x_t)$ (Song et al., 2021b)). In particular, the conditional score for a fixed $y$ can be expressed as the sum of the unconditional score and the score of the classifier, that is, $\nabla_{x_t} \log p_t(x_t|y) = \nabla_{x_t} \log p_t(x_t) + \nabla_{x_t} \log p_t(y|x_t)$. We take motivation from an alternative way to allow for controllable generation, which, given supervised samples $(x, y(x))$, uses the following training objective for each time $t$

$$J_t^{CSM}(\theta) = \mathbf{E}_{x_0} \{ \mathbf{E}_{x_t|x_0} [\|s_\theta(x_t, t, y(x_0)) - \nabla_{x_t} \log p_{0t}(x_t|x_0)\|_2^2] \}. \tag{5}$$

The objective in equation 5 is minimized if and only if the model equals the conditional score function $\nabla_{x_t} \log p_t(x_t|y(x_0) = \hat{y})$ for all labels $\hat{y}$.

## 2 Diffusion-based Representation Learning

We begin this section by presenting an alternative formulation of the Denoising Score Matching (DSM) objective, which shows that this objective cannot be made arbitrarily small. Formally, the formula of the DSM objective can be rearranged as

$$J_t^{DSM}(\theta) = \mathbf{E}_{x_0} \{ \mathbf{E}_{x_t|x_0} \left[ \|\nabla_{x_t} \log p_{0t}(x_t|x_0) - \nabla_{x_t} \log p_t(x_t)\|_2^2 + \|s_\theta(x_t, t) - \nabla_{x_t} \log p_t(x_t)\|_2^2 \right] \}. \tag{6}$$

The above formulation holds, because the DSM objective in equation 4 is minimized when $\forall x_t : s_\theta(x_t, t) = \nabla_{x_t} \log p_t(x_t)$, and differs from ESM in equation 3 only by a constant (Vincent, 2011). Hence, the constant is equal to the minimum achievable value of the DSM objective. A detailed proof is included in the Appendix B.

It is noteworthy that the first term in the right-hand side of the equation 6 does not depend on the learned score function of $x_t$ for every $t \in [0, T]$. Rather, it is influenced by the diffusion process that generates $x_t$ from $x_0$. This observation has not been emphasized previously, probably because it has no direct effect on the learning of the score function, which is handled by the second term in the equation 6. However, the additional constant has major implications for finding other hyperparameters such as the function $\lambda(t)$ and the choice of $\sigma(t)$ in the forward SDE. As Kingma et al. (2021) shows, changing the integration variable from time to signal-to-noise ratio (SNR) simplifies the diffusion loss such that it only depends on the end values of SNR. Hence, the loss is invariant to the intermediate values of the noise schedule. However, the weight functions $\lambda(\cdot)$ is still an important hyper-parameter whose choice might be affected by the non-vanishing constant in Equation 6.

To the best of our knowledge, there is no known theoretical justification for the values of $\sigma(t)$. While these hyperparameters could be optimized in ESM using gradient-based learning, this ability is severely limited by the non-vanishing constant in equation 6.

Even though the non-vanishing constant in the denoising score matching objective presents a burden in multiple ways such as hyperparameter search and model evaluation, it provides an opportunity for latent representation learning, which will be described in the following sections. We note that this is different from Sinha et al. (2021); Mittal et al. (2021) as they consider a Variational Autoencoder model followed by diffusion in the latent space, where their representation learning objective is still guided by reconstruction. Contrary to this, our representation learning approach does not utilize a variational autoencoder model and is guided by denoising instead. Our approach is similar to Preechakul et al. (2021) but we also condition the encoder system on the time-step, thereby improving representation capacity and leading to parameterized curve-based representations.

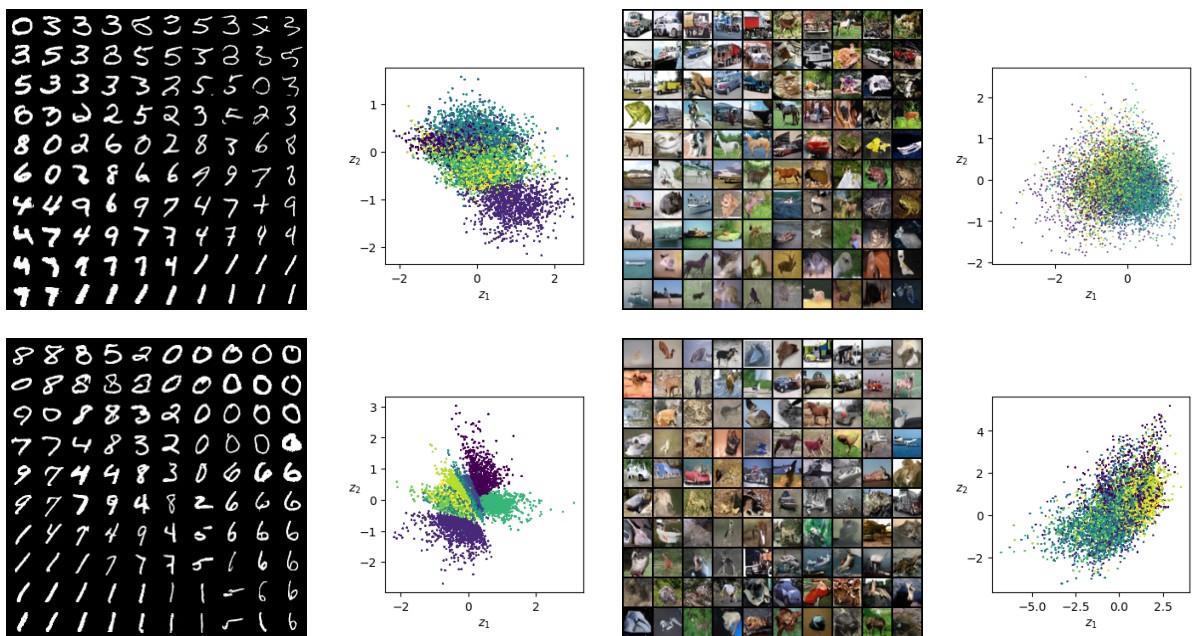

Figure 2: Results of proposed DRL models trained on MNIST and CIFAR-10 with point clouds visualizing the latent representation of test samples, colored according to the digit class. The models are trained with **Top:** uniform sampling of $t$ and **Bottom:** a focus on high noise levels. Samples are generated from a grid of latent values ranging from -1 to 1.

## 2.1 Learning latent representations

Since supervised data is limited and rarely available, we propose to learn a labeling function $y(x_0)$ at the same time as optimizing the conditional score matching objective in equation 5. In particular, we represent the labeling function as a trainable encoder $E_\phi : \mathbb{R}^d \to \mathbb{R}^c$, where $E_\phi(x_0)$ maps the data sample $x_0$ to its corresponding code in the $c$-dimensional latent space. The code is then used as additional input to the score model. Formally, the proposed learning objective for Diffusion-based Representation Learning (DRL) is the following:

$$J^{DRL}(\theta, \phi) = \mathbf{E}_{t,x_0,x_t}[\lambda(t)\|s_\theta(x_t, t, E_\phi(x_0)) - \nabla_{x_t} \log p_{0t}(x_t|x_0)\|_2^2] \tag{7}$$

To get a better idea of the above objective, we provide an intuition for the role of $E_\phi(x_0)$ in the input of the model. The model $s_\theta(\cdot, \cdot, \cdot) : \mathbb{R}^d \times \mathbb{R} \times \mathbb{R}^c \to \mathbb{R}^d$ is a vector-valued function whose output points to different directions based on the value of its third argument. In fact, $E_\phi(x_0)$ selects the direction that best recovers $x_0$ from $x_t$. Hence, when optimizing over $\phi$, the encoder learns to extract the information from $x_0$ in a reduced-dimensional space that helps recover $x_0$ by denoising $x_t$.

We show in the following that equation 7 is a valid representation learning objective. The score of the perturbation kernel $\nabla_{x_t} \log p_{0t}(x_t|x_0)$ is a function of only $t$, $x_t$ and $x_0$. Thus, the objective can be reduced to zero if all information about $x_0$ is contained in the latent representation $E_\phi(x_0)$. When $E_\phi(x_0)$ has no mutual information with $x_0$, the objective can only be reduced up to the constant in equation 6. Hence, our proposed formulation takes advantage of the non-zero lower-bound of equation 6, which can only vanish when the encoder $E_\phi(\cdot)$ properly distills information from the unperturbed data into a latent code, which is an additional input to the score model. These properties show that equation 7 is a valid objective for representation learning.

Our proposed representation learning objective enjoys the continuous nature of SDEs, a property that is not available in many previous representation learning methods (Radford et al., 2016; Chen et al., 2016; Locatello et al., 2019). In DRL, the encoder is trained to represent the information needed to denoise $x_0$ for different levels of noise $\sigma(t)$. We hypothesize that by adjusting the weighting function $\lambda(t)$, we can manually control

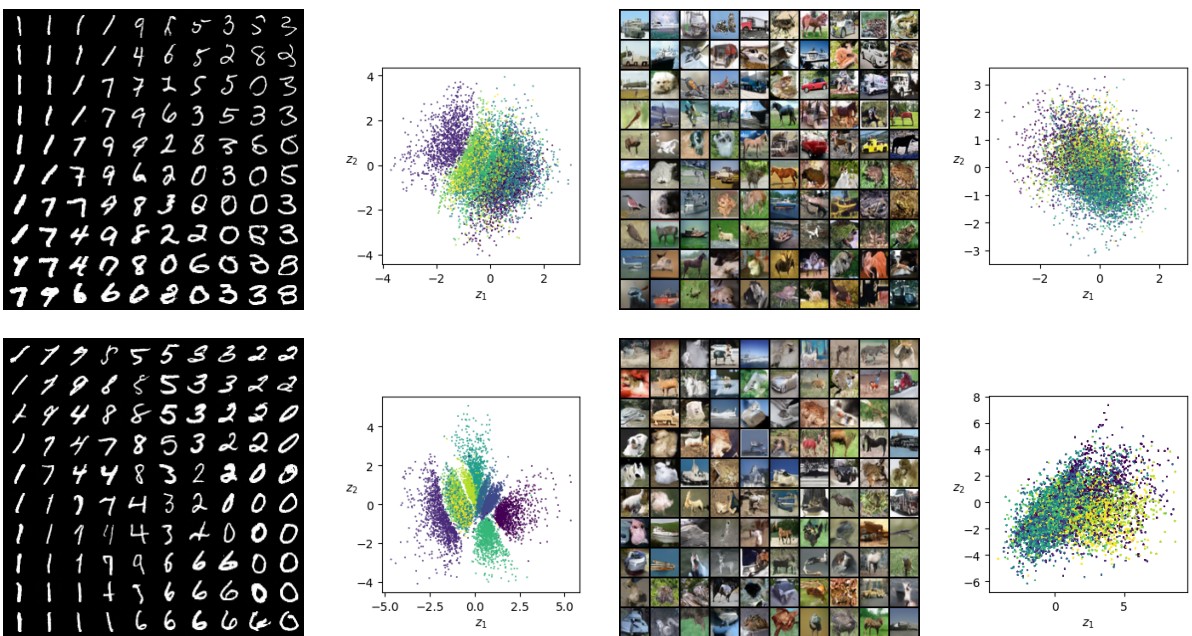

Figure 3: Results of proposed VDRL models trained on MNIST and CIFAR-10 with point clouds visualizing the latent representation of test samples, colored according to the digit class. The models are trained with **Top:** uniform sampling of $t$ and **Bottom:** a focus on high noise levels. Samples are generated from a grid of latent values ranging from -2 to 2.

the granularity of the features encoded in the representation and provide empirical evidence as support. Note that $t \to T$ is associated with higher levels of noise and the mutual information of $x_t$ and $x_0$ starts to vanish. In this case, denoising requires all information about $x_0$ to be contained in the code. In contrast, $t \to 0$ corresponds to low noise levels and hence $x_t$ contains coarse-grained features of $x_0$ and only fine-grained properties may have been washed out. Hence, the encoded representation learns to keep the information needed to recover these fine-grained details. We provide empirical evidence to support this hypothesis in Section 3.

It is noteworthy that $E_\phi$ does not need to be a deterministic function and can be a probabilistic map similar to the encoder of VAEs. In principle, it can be viewed as an information channel that controls the amount of information that the diffusion model receives from the initial point of the diffusion process. With this perspective, any deterministic or stochastic function that can manipulate $I(x_t, x_0)$, the mutual information between $x_0$ and $x_t$, can be used. This opens up the room for stochastic encoders similar to VAEs which we call Variational Diffusion-based Representation Learning (VDRL). The formal objective of VDRL is

$$J^{VDRL}(\theta, \phi) = \mathbf{E}_{t,x_0,x_t}[\mathbf{E}_{z \sim E_\phi(Z|x_0)}[\lambda(t)\|s_\theta(x_t, t, z) - \nabla_{x_t} \log p_{0t}(x_t|x_0)\|_2^2] \\ + \mathcal{D}_{\mathrm{KL}}(E_\phi(Z|x_0)||\mathcal{N}(Z; 0, I)]$$
(8)

## 2.2 Infinite-dimensional representation of data

We now present an alternative version of DRL where the representation is a function of time. Instead of emphasizing on different noise levels by weighting the training objective, as done in the previous section, we can provide the time $t$ as input to the encoder. Formally, the new objective is

$$\mathbf{E}_{t,x_0,x_t}[\lambda(t)\|s_\theta(x_t, t, E_\phi(x_0, t)) - \nabla_{x_t} \log p_{0t}(x_t|x_0)\|_2^2]$$
(9)

where $E_\phi(x_0)$ in equation 7 is replaced by $E_\phi(x_0, t)$. Intuitively, it allows the encoder to extract the necessary information of $x_0$ required to denoise $x_t$ for any noise level. This leads to richer representation learning since normally in autoencoders or other *static* representation learning methods, the input data $x_0 \in \mathbb{R}^d$ is mapped to a single point $z \in \mathbb{R}^c$ in the latent space. In contrast, we propose a richer representation where the input

$x_0$ is mapped to a curve in $\mathbb{R}^c$ instead of a single point. Hence, the learned latent code is produced by the map $x_0 \rightarrow (E_\phi(x_0, t))_{t \in [0,T]}$ where the infinite-dimensional object $(E_\phi(x_0, t))_{t \in [0,T]}$ is the encoding for $x_0$.

**Proposition 1.** *For any downstream task, the infinite-dimensional code $(E_\phi(x_0, t))_{t \in [0,T]}$ learned using the objective in equation 9 is at least as good as finite-dimensional static codes learned by the reconstruction of $x_0$.*

*Proof sketch.* Let $L_D(z, y)$ be the per-sample loss for a supervised learning task where z(x) which is calculated for the pair (z, y) where z = z(x, t) is the representation learned for the input x and y is the label. The representation function is also a function of the scalar $t$ that takes values from a closed subset $U$ of $R$. For any value $s \in U$, it is obvious that

$$min_{t \in U} L_D(z(x, t), y) < L_D(z(x, s), y) \tag{10}$$

Taking into account the extra argument $t$, the representation function $z(x, t)$ can be seen as an infinite dimensional representation. The argument $t$ actually controls which representation of $x$ has to be passed to the downstream task. The conventional representation learning algorithms correspond to choosing the $t$ argument apriori and keep it fixed independent of $x$. Here, by minimizing over $t$, the passed representation cannot be worse than the results of conventional representation learning methods.

The score matching objective can be seen as a reconstruction objective of $x_0$ conditioned on $x_t$. The terminal time $T$ is chosen large enough so that $x_T$ is independent of $x_0$, hence the objective for $t = T$ is equal to a reconstruction objective without conditioning. Therefore, there exists a $t \in [0, T]$ where the learned representation $E_\phi(x_0, t)$ is the same representation learned by the reconstruction objective of a vanilla autoencoder. The full proof for Proposition 1 can be found in the Appendix C

A downstream task can leverage this rich encoding in various ways, including the use of either the static code for a fixed $t$, or the use of the whole trajectory $(E_\phi(x_0, t))_{t \in [0,T]}$ as input. We posit the conjecture that the proposed rich representation is helpful for downstream tasks when used for pretraining, where the value of $t$ could either be a model selection parameter or be jointly optimized with other parameters during training. We leave investigations along these directions as important future work. We show the performance of the proposed model on downstream tasks in Section 3.1 and also evaluate it on semi-supervised image classification in Section 3.2.

## 3 Results

For all experiments, we use the same function $\sigma(t), t \in [0, 1]$ as in Song et al. (2021b), which is $\sigma(t) = \sigma_{\min} (\sigma_{\max}/\sigma_{\min})^t$, where $\sigma_{\min} = 0.01$ and $\sigma_{\max} = 50$. Further, we use a 2d latent space for all qualitative experiments (Section 3.3) and 128 dimensional latent space for the downstream tasks (Section 3.1) and semi-supervised image classification (Section 3.2). We also set $\lambda(t) = \sigma^2(t)$, which has been shown to yield the KL-Divergence objective (Song et al., 2021a). Our goal is not to produce state-of-the-art image quality, rather showcase the representation learning method. Because of that and also limited computational resources, we did not carry out an extensive hyperparameter sweep (check Appendix D for details). Note that all experiments were conducted on a single RTX8000 GPU, taking up to 30 hours of wall-clock time, which only amounts to 15% of the iterations proposed in Song et al. (2021b).

### 3.1 Downstream Classification

We directly evaluate the representations learned by different algorithms on downstream classification tasks for CIFAR10, CIFAR100, and Mini-ImageNet datasets. The representation is first learned using the proposed diffusion-based method. Then, the encoder (either deterministic or probabilistic) is frozen and a single-layered neural network is trained on top of it for the downstream prediction task. For the baselines, we consider an Autoencoder (AE), a Variational Autoencoder (VAE), and a restricted ~~handicapped~~ Contrastive Learning (SimCLR-Gauss explained below) setup to compare with the proposed methods (DRL and VDRL). Figure 4 shows that DRL and VDRL outperform autoencoder-styled baselines as well as the restricted ~~handicapped~~ contrastive learning baseline.

*Restricted ~~Handicapped~~ simCLR—* To obtain a fair comparison, we restricted the transformations used by the simCLR method to the additive pixel-wise Gaussian noise (SimCLR-Gauss). The original simCLR expectedly outperforms the other methods because it uses the privileged information injected by the employed data

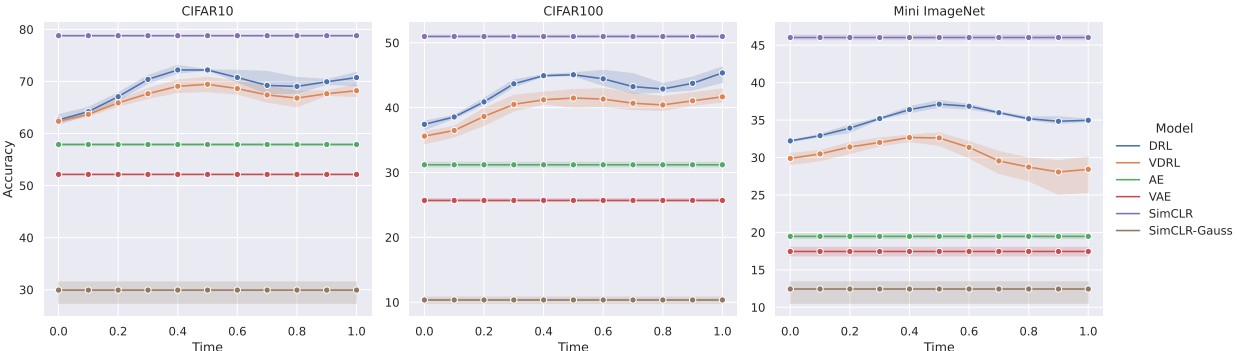

Figure 4: Comparing the performance of the proposed diffusion-based representations (DRL and VDRL) with the baselines that include autoencoder (AE), variational autoencoder (VAE), simple contrastive learning (simCLR) and its restricted ~~handicapped~~ variant (simCLR-Gauss) which exclude domain-specific data augmentation from the original simCLR algorithm.

| Pretraining Mixup | | LaplaceNet None No | LaplaceNet None Yes | Ours DRL No | Ours DRL Yes | Ours VDRL No |
|---|---|---|---|---|---|---|
| Dataset | #labels | | | | | |
| CIFAR-10 | 100 | 73.68 | 75.29 | 74.31 | 64.67 | **81.63** |
| | 500 | 91.31 | 92.53 | **92.70** | 92.31 | **92.79** |
| | 1000 | 92.59 | 93.13 | **93.24** | **93.42** | **93.60** |
| | 2000 | 94.00 | 93.96 | **94.18** | 93.91 | 93.96 |
| | 4000 | 94.73 | 94.97 | 94.75 | **95.22** | **95.00** |
| CIFAR-100 | 1000 | 55.58 | 55.24 | **55.85** | 55.74 | **56.47** |
| | 4000 | 67.07 | 67.25 | 67.22 | **67.47** | **67.54** |
| | 10000 | 73.19 | 72.84 | **73.31** | **73.66** | **73.50** |
| | 20000 | 75.80 | 76.07 | **76.46** | **76.88** | **76.64** |
| MiniImageNet | 4000 | 58.40 | 58.84 | **58.95** | **59.29** | **59.14** |
| | 10000 | 66.65 | 66.80 | **67.31** | 66.63 | **67.46** |

Table 1: Comparison of classifier accuracy in % for different pretraining settings. Scores better than the SOTA model (LaplaceNet) are in **bold**. "DRL" pretraining is our proposed representation learning, and "VDRL" the respective version which uses a probabilistic encoder.

augmentation methods. For example, random cropping is an inductive bias that reflects the spatial regularity of the images. Even though it is possible to strengthen our method and autoencoder-based baselines such as VAEs with such augmentation-based strategies, we restricted all baselines to the generic setting without this inductive bias and leave the domain-specific improvements for future work.

It is seen that the DRL and VDRL methods significantly outperform the baselines on all the datasets at a number of different time-steps $t$. We further evaluate the infinite-dimensional representation on few-shot image classification using the representation at different timescales as input. The detailed results are shown in Appendix E. In summary, the representations of DRL and VDRL achieve significant improvements as compared to an autoencoder or VAE for several values of $t$ .

Overall the results align with the theoretical argument of Proposition 1 that the rich representation of DRL is at least as good as the static code learned using a reconstruction objective. It further shows that in practice, the infinite-dimensional code is superior to the static (finite-dimensional) representation for downstream applications such as image classification by a significant margin.

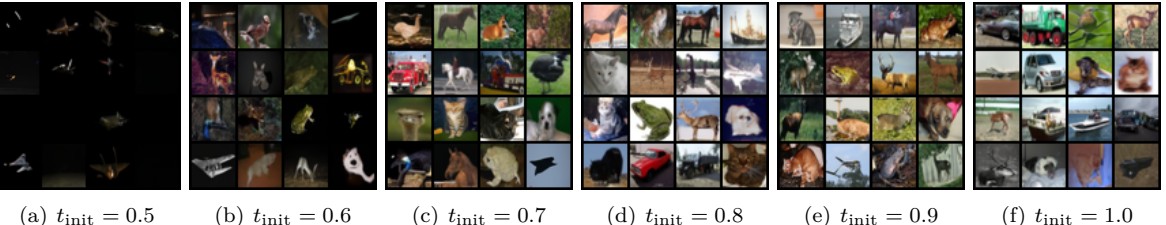

(a) $t_{\text{init}} = 0.5$     (b) $t_{\text{init}} = 0.6$     (c) $t_{\text{init}} = 0.7$     (d) $t_{\text{init}} = 0.8$     (e) $t_{\text{init}} = 0.9$     (f) $t_{\text{init}} = 1.0$

Figure 5: Generated image samples for different values of $t_{\text{init}}$ using the Gaussian prior.

### 3.2 Semi-Supervised Image Classification

The current state-of-the-art model for many semi-supervised image classification benchmarks is LaplaceNet (Sellars et al., 2021). It alternates between assigning pseudo-labels to samples and supervised training of a classifier. The key idea is to assign pseudo-labels by minimizing the graphical Laplacian of the prediction matrix, where similarities of data samples are calculated on a hidden layer representation in the classifier. Note that LaplaceNet applies *mixup* (Zhang et al., 2017) that changes the input distribution of the classifier. We evaluate our method with and without mixup on CIFAR-10 (Krizhevsky et al., a), CIFAR-100 (Krizhevsky et al., b) and MiniImageNet (Vinyals et al., 2016).

In the following, we evaluate the infinite-dimensional representation $(E_\phi(x_0, t))_{t \in [0,T]}$ on semi-supervised image classification, where we use DRL and VDRL as pretraining for the LaplaceNet classifier. Table 1 depicts the classifier accuracy on test data for different pretraining settings. Details for architecture and hyperparameters are described in Appendix G.

Our proposed pretraining using DRL significantly improves the baseline and often surpasses the state-of-the-art performance of LaplaceNet. Most notable are the results of DRL and VDRL without mixup, which achieve high accuracies without being specifically tailored to the downstream task of classification. Note that pretraining the classifier as part of an autoencoder did not yield any improvements (Table 4 in the Appendix). Combining DRL with mixup yields inconsistent improvements, results are reported in Table 5 of the Appendix. In addition, DRL pretraining achieves much better performances when only limited computational resources are available (Tables 2, 3 in the Appendix).

### 3.3 Qualitative Results

We first train a DRL model with $L_1$-regularization on the latent code on MNIST (LeCun & Cortes, 2010) and CIFAR-10. Figure 2 (*top*) shows samples from a grid over the latent space and a point cloud visualization of the latent values $z = E_\phi(x_0)$. For MNIST, we can see that the value of $z_1$ controls the stroke width, while $z_2$ weakly indicates the class. The latent code of CIFAR-10 samples mostly encodes information about the background color, which is weakly correlated to the class. The use of a probabilistic encoder (VDRL) leads to similar representations, as seen in Fig. 3 (*top*). We further want to point out that the generative process using the reverse SDE involves randomness and thus generates different samples for a single latent representation. The diversity of samples however steadily decreases with the dimensionality of the latent space, shown in Figure 7 of the Appendix.

Next, we analyze the behavior of the representation when adjusting the weighting function $\lambda(t)$ to focus on higher noise levels, which can be done by changing the sampling distribution of $t$. To this end, we sample $t \in [0, 1]$ such that $\sigma(t)$ is uniformly sampled from the interval $[\sigma_{\min}, \sigma_{\max}] = [0.01, 50]$. Figure 2 (*bottom*) shows the resulting representation of DRL and Figure 3 (*bottom*) for the VDRL results. As expected, the latent representation for MNIST encodes information about classes rather than fine-grained features such as stroke width. This validates our hypothesis of Section 2.1 that we can control the granularity of features encoded in the latent space. For CIFAR-10, the model again only encodes information about the background, which contains the most information about the image class. A detailed analysis of class separation in the extreme case of training on single timescales is included in Appendix H.

Overall, the difference in the latent codes for varying $\lambda(t)$ shows that we can control the granularity encoded in the representation of DRL. This ability provides a significant advantage when there exists some prior

information about the level of detail that we intend to encode in the target representation. We further illustrate how the representation encodes information for the task of denoising in the Appendix (Fig. 6).

We also provide further analysis into the impact of noise scales on generation in Appendix I.

## 4 Conclusion

We presented Diffusion-based Representation Learning (DRL), a new objective for representation learning based on conditional denoising score matching. In doing so, we turned the original non-vanishing objective function into one that can be reduced arbitrarily close to zero by the learned representation. We showed that the proposed method learns interpretable features in the latent space. In contrast to some of the previous approaches that required specialized architectural changes or data manipulations, denoising score matching comes with a natural ability to control the granularity of features encoded in the representation. We demonstrated that the encoder can learn to separate classes when focusing on higher noise levels and encodes fine-grained features such as stroke-width when mainly trained on smaller noise variance. In addition, we proposed an infinite-dimensional representation and demonstrated its effectiveness for downstream tasks such as few-shot classification. Using the representation learning as pretraining for a classifier, we were able to improve the results of LaplaceNet, a state-of-the-art model on semi-supervised image classification.

Starting from a different origin but conceptually close, contrastive learning as a self-supervised approach could be compared with our representation learning method. We should emphasize that there are fundamental differences both at theoretical and algorithmic levels between contrastive learning and our diffusion-based method. The generation of positive and negative examples in contrastive learning requires the domain knowledge of the applicable invariances. This knowledge might be hard to obtain in scientific domains such as genomics where the knowledge of invariance amounts to the knowledge of the underlying biology which in many cases is not known. However, our diffusion-based representation uses the natural diffusion process that is employed in score-based models as a continuous obfuscation of the information content. Moreover, unlike the loss function of the contrastive-based methods that are specifically designed to learn the invariances of manually augmented data, our method uses the same loss function that is used to learn the score function for generative models. The representation is learned based on a generic information-theoretic concept which is an encoder (information channel) that controls how much information of the input has to be passed to the score function at each step of the diffusion process. We also provided theoretical motivation for this information channel. The algorithm cannot ignore this source of information because it is the only way to reduce a non-negative loss arbitrarily close to zero.

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
