# OpenReview forum: "Diffusion Based Representation Learning"
_TMLR — Rejected by TMLR_

### Review · Reviewer_c45b · 2022-07-24

**Summary Of Contributions:**

This work proposes to re-purpose the loss function of diffusion models for representation learning. The time-conditional score function is modified to incorporate a learnable embedding function so that non-negative denoising score matching losses may be made arbitrarily close to zero. Authors demonstrate that representations from this embedding function outperform those obtained by autoencoders, VAEs and handicapped simCLR. Moreover, these representations can be generalized to be non-deterministic via a VAE framework, and to infinite dimensions when additionally conditioned on time.

**Broader Impact Concerns:**

No particular concerns from my understanding.

**Requested Changes:**

I suggest authors thinking carefully about potential changes below:

1. Include prior works on denoising autoencoders for representation learning as additional baselines in experiments.

2. Clarify the importance of VDRL, and motivate the formal objective provided in equation (8).

3. Make Proposition 1 rigorous.

4. Regenerate figures in Section 3.3 with probability flow ODE or reverse SDE with a shared random seed.

5. Bring results on infinite dimensional representations from appendix to the main paper, and clarify how to substitute infinite dimensional codes for finite dimensional ones.

**Strengths And Weaknesses:**

### Strengths:

The proposed approach is simple to understand and easy to use. The paper is well-written.

### Weaknesses:

1. Representation learning with multi-scale denoising score matching has been proposed before in [1][2][3]. Although authors cited them, it is unknown how the proposed DRL/VDRL fares against existing methods experimentally.

2. It is unclear what advantages VDRL brings to the table, since VDRL is strictly dominated by DRL in Figure 4. Moreover, the VDRL objective is not well-motivated. Equation (8) is not an ELBO unless you can write the reconstruction loss as the negative log-likelihood of some decoder. What is the form of this decoder?

3. Proposition 1 is not a rigorous statement. What's the technical definition of one code being "at least as good as" another code? A proposition shouldn't contain vague statements which you cannot prove.

4. In Section 3.3, authors complain that "the generative process using the reverse SDE involves randomness and thus generates different samples for a single latent representation". This can be mitigated by either fixing the random seed for each SDE generation, or using the probability flow ODE.

5. It is unclear how the infinite dimensional latent representation can be used for representation learning, and what the results are in the main part of this paper.

References:

[1] K. J. Geras and C. Sutton. Scheduled denoising autoencoders. arXiv preprint arXiv:1406.3269, 2014.

[2] B. Chandra and R. K. Sharma. Adaptive noise schedule for denoising autoencoder. NeurIPS 2014.

[3] Q. Zhang and L. Zhang. Convolutional adaptive denoising autoencoders for hierarchical feature extraction. Frontiers of Computer Science, 2018.

---

> ### Author Response · Authors · 2022-10-02
> **Response to Reviewer**
>
> **Connections to Prior Work**: We thank the reviewer for their concerns. Since the cluster we have been using was heavily occupied for the ICLR deadline, we could not run additional experiments. However, we will be working on getting a vanilla denoising autoencoder baseline for additional comparison.
>
> **Advantages of VDRL**: We propose VDRL and DRL as two prospective ways of performing representation learning using the diffusion-based generative modeling paradigm. The two approaches differ just in the regularization used on the latent code, and our aim here was to highlight that with two different choices of regularizations and the choice of stochastic vs deterministic encoder, the model is able to perform representation learning.
>
> We thank the reviewer for pointing out that Equation (8) is not a valid ELBO. We agree, and clarify that the use of KL was solely as a regularizer and that the whole objective does not constitute a generative model but is instead a way to augment the score-matching generative modeling approach to obtain representations from data samples.
>
> **Proposition 1**: We clarify the proposition here, and also provide this in the revised version of the draft.
>
> Let $L_D(z, y)$ be the per-sample loss for a supervised learning task where z(x) which is calculated for the pair (z, y) where z = z(x, t) is the representation learned for the input x and y is the label. The representation function is also a function of the scalar $t$ that takes values from a closed subset $U$ of $R$. For any value  $s\in U$, it is obvious that
>
> $min_{t\in U} L_D(z(x, t), y) < L_D(z(x, s), y)$.
>
> Taking into account the extra argument $t$, the representation function $z(x, t)$ can be seen as an infinite dimensional representation. The argument $t$ actually controls which representation of $x$ has to be passed to the downstream task. The conventional representation learning algorithms correspond to choosing the $t$ argument apriori and keep it fixed independent of $x$. Here, by minimizing over $t$, the passed representation cannot be worse than the results of conventional representation learning methods.
>
> **Randomness in Sampling**: We do note claim that generating different samples from the same latent representation is a problem. Our aim here was to point out that typical diffusion based models have randomness associated with the sampling procedure, and that the representation learning objective learns a meaningful representation because it steadily reduces the randomness of the samples obtained when conditioned on the same code.
>
> **Advantages of Infinite-Dimensional Latent Representation**: While the approach proposed in this work leads to an infinite-dimensional representation for each data sample, we believe that the study of the semantics and properties of such an infinite dimensional representation is out of scope of this work. Our work showcases the representations learned using this approach (Figures 2 and 3), the performance of this representation obtained at different points in the trajectory, as highlighted in Figure 4, as well as usage of this approach for semi-supervised image classification in Table 1. There has already been some work done on the analysis of the infinite-dimensional representations obtained from this setup in Mittal et. al 2022.
>
> *Mittal et. al 2022: From Points to Functions: Infinite dimensional Representations in Diffusion Models*

---

### Review · Reviewer_MPuM · 2022-08-17

**Summary Of Contributions:**

This paper proposes a representation learning framework based on diffusion-based generative models. Specifically, the authors introduce an extra trainable encoder and condition the score model on the output of the encoder. The authors train the model using a proposed objective that shares similarity with the diffusion model objective except that the loss is now conditioned on the output of the encoder. The authors further introduce another variational objective that is similar to stacking a series of denoising autoencoders. The authors show that the proposed approach can achieve reasonable performance on representation learning.

**Broader Impact Concerns:**

The current paper does not have enough discussion about the potential negative societal impact.

**Requested Changes:**

1. Writing: The presentation of "Introduction" can be improved. Currently, there is too much background and prior work, and the contribution of the work is only briefly introduced at the very last paragraph. It would be good to introduce your approach more before spending too much text talking about the background. The formulation of "proposition 1" will also need to be improved: what does "at least as good as" mean? Good in terms of what metric? It is not very convincing that the claim holds for any downstream task, more explanations and justifications are needed.

2.  [1, 3] are relevant works. It would be good to also compare with them. In particular, [1] is very similar to this work. It would be good if the authors could clarify the differences.

3. Stronger baselines and more convincing results (see weakness).


[1] https://arxiv.org/abs/2111.15640

[2] https://arxiv.org/abs/2103.16091

[3] https://arxiv.org/abs/2106.06819

**Strengths And Weaknesses:**

**Strength:**
1. Incorporating representation learning into generative modeling is a very interesting and natural idea. (However, a drawback is that it might cause the model to achieve not very impressive performance on both tasks.)

2. The presentation of the paper is relatively clear.

**Weakness:**
1. The novelty can be limited. Incorporating an extra encoder into a diffusion model is not a new idea. For instance, [1, 2, 3]  have already introduced extra encoders in the latent space of a diffusion model, and condition the diffusion model on the output of the encoder. Specifically, [1, 3] have also shown that such approaches can learn useful representation which can later be used for conditional generation and image editing. The variational objective function also looks like stacking a series of denoising autoencoders.

2.  Baselines are relatively weak and not convincing. It is known that data augmentation (e.g., cropping, color jittering) plays an important role in contrastive learning frameworks. It might not be a fair comparison to consider a simCLR framework using only Gaussian noise as data augmentation. Instead, a more proper comparison would be to compare with simCLR in a default setting and use the same data augmentation (or only Gaussian noise augmentation) for the proposed diffusion framework.

3. Missing baselines. [1, 3] are relevant works. It would be good to also compare with them and add discussions accordingly. In particular, [1] is very similar to this work. It would be good if the authors could clarify the differences.

4. The performances are not very impressive. For instance, in Table1, the proposed approach only has very marginal improvements when #labels>=500. The error rates are also not reported: how many runs are there? In Fig4, it seems that SimCLR can easily outperform the proposed approach. Adding extra data augmentation is easy for SimCLR, but might hinder the generation performance for the proposed approach. It is not convincing to simply compare with SimCLR-Gauss. Instead, the authors can consider incorporating more data agumentations into the current diffusion framework and compare directly with (default) state-of-the-art self-supervised representation learning algorithms.

5. Since the authors show the samples, it would be good to compare with regular diffusion models and see how the FID/IS changes after introducing an encoder and trained using the proposed objective.

6. The writing can be improved and some terms need to be defined. For instance, in "proposition 1", "at least as good as" is very vague. Also see "requested changes".

[1] https://arxiv.org/abs/2111.15640

[2] https://arxiv.org/abs/2103.16091

[3] https://arxiv.org/abs/2106.06819

---

> ### Author Response · Authors · 2022-10-02
> **Response to Reviewer**
>
> **Novelty**: We thank the reviewer for bringing these works to our attention, and we have revised our draft to include them in our discussions. [3] aims to learn a Variational Autoencoder, where the latent space is then modeled through a diffusion model. The representations for each sample are still, however, learned through the encoding/decoding process, which is in contrast to our approach as the representations are learned through an additional input to the score model, and hence is strongly tied with denoising in the pixel space itself. Further, for their few-shot settings, they do assume access to some properties or features whereas our method, as presented, is completely unsupervised. [2] is also similar to [3] in the sense that they use a VAE-model to learn the latent representation, which then undergoes a diffusion process. In contrast, our representation learning is guided by denoising objective as opposed to reconstruction, and we do not make use of the VAE objectives.
>
> [1] is quite similar, and concurrent, to our work. While they focus on disentanglement, interpolation and generating samples, we put our focus on downstream tasks like image classification once the representation is learned. They also use a secondary DDIM process to allow for unconditional generation. We, on the other hand, consider the problem of representation learning for downstream predictions and thus do not care about generation in this setup and extend the settings provided in [1] by allowing for unbounded representation space using a trajectory / curve representation for every sample. We would like to stress that [1] is concurrent to our work, and we provide additional capacity of learning infinite-dimensional representations for samples.
>
> **Weak Baselines**: We point the readers to Figure 4 which contains comparison with the default SimCLR setting as well. We also saw that adding transformations to DRL or VDRL didn’t help much. We would like to highlight that using the same augmentations as SimCLR in DRL/VDRL, or even autoencoders, is not a fair comparison because in SimCLR there is the added supervision or knowledge about when semantic information is retained (via transformations) and when it is not (random pairing). This additional supervision is the prominent reason behind the workings of SimCLR and it is hard to bake that in without a contrastive component. Hence, we believe that even with the same comparisons, SimCLR has additional supervision by default and thereby not directly comparable with the proposed methods.
>
> **FID/IS Comparison**: We would like to take this opportunity to emphasize that the proposed model is not a generative model but instead is a methodology that uses diffusion-based generative models for representation learning. Given that it is not a generative model, we believe that FID/IS score is not a valid metric for the proposed system.
>
> **Proposition 1**: Let $L_D(z, y)$ be the per-sample loss for a supervised learning task where z(x) which is calculated for the pair (z, y) where z = z(x, t) is the representation learned for the input x and y is the label. The representation function is also a function of the scalar $t$ that takes values from a closed subset $U$ of $R$. For any value  $s\in U$, it is obvious that
>
> $min_{t\in U} L_D(z(x, t), y) < L_D(z(x, s), y)$.
>
> Taking into account the extra argument $t$, the representation function $z(x, t)$ can be seen as an infinite dimensional representation. The argument $t$ actually controls which representation of $x$ has to be passed to the downstream task. The conventional representation learning algorithms correspond to choosing the $t$ argument apriori and keep it fixed independent of $x$. Here, by minimizing over $t$, the passed representation cannot be worse than the results of conventional representation learning methods.

---

### Review · Reviewer_o2f5 · 2022-09-07

**Summary Of Contributions:**

This paper presents a method for learning representations with diffusion models. By using conditional denoising score matching combined with a learned encoder, the proposed approach is able to learn a low-dimensional representation of the clean input image that performs well on downstream tasks. The method is extended to learn a time-dependent representation of the clean input data that the score-based model is conditioned on. At test-time the time scalar can be optimized or searched over to idetnify the representation, and the weighting on different levels in the diffusion model can be leveraged to control the granularity of the representation. The method is applied to unsupervised and semi-supervised representation learning on CIFAR-10, CIFAR-100, and Mini ImageNet, where it often outperforms prior work when restricted to permutation-invariant augmentations (additive Gaussian noise).


**Broader Impact Concerns:**

Broader impact section was not present, but I did not find this work raided any immediate ethical concerns other than a term that I requested to be changed.

**Requested Changes:**



Requested Changes:
* Experimental: Running experiments on ImageNet-1K really feels like a necessity for representation learning methods these days, as many recent papers do not evaluate on CIFAR-10 or CIFAR-100. I do also think it is fair to compare your approach to recent methods like Masked Autoencoders (https://arxiv.org/abs/2111.06377) as randomly dropping patches doesn’t feel like stronger knowledge/augmentation than using convolutions. Both are already taking advantage of the spatial structure of the input. If you cannot run on ImageNet, then maybe consider trying out a MAE-like approach to compare to on MiniImageNet?
* Clarifying what the time-dependent encoder is doing and how that is related to the notion of learning a generative model of the input data
* Do not use the word handicap to describe SimCLR without standard augmentations, just say SimCLR-Gauss or Restricted SimCLR
* Fig 2 and 3 look basically the same, and in general the DRL and VDRL results aren’t very different. The text describing figures is often very far from the figure itself, please try to improve the layout.
* Infinite-dimensional code: not a required change, but I find using the term “infinite-dimensional code” to describe a finite-dimensional code indexed by a continuous variable (t) misleading. Typically I think of infinite-dimensional representations as relating to non-parametric methods.
* Equation 6 / Appendix B on DSM is just denoising score matching applied to the conditional distribution p(x|y) instead of an unconditional p(x). The appendix of Vincent (2011) similarly has the exact constants relating DSM to ESM, but you’ve lumped their C_2 + C_3 into one constant c. Is there something distinct? I think it’s clearer if you present it as DSM applied to the conditional vs. arguing it’s a new and different objective.
* Why do surprised features for large timesteps work well? Given that optimal denoiser at high noise levels just gives back the mean image for the class
* No error bars in Table 1, hard to know what is/is not a significant difference. For exmaple, across different subets of 100 supervised labels, what’s the variance in performance?
* While contrastive learning encourages invariance through augmentation, simply using the representation one layer before the readout head often contains that information (see e.g. SimCLR)
* Might be worth citing/discussing relationship between conditional score matching and prior work like AR score matching (matching conditional distributions w/DSM): https://proceedings.neurips.cc/paper/2020/file/4a4526b1ec301744aba9526d78fcb2a6-Supplemental.pdf
* “No known theoretical justification for sigma(t)”: https://arxiv.org/abs/2107.00630 shows that for a continuous-time diffusion loss the shape of the schedule does not matter for the mean loss only the endpoints. But it does impact variance and they motivate a variance-minimization scheme for the schedule
* “single-layered neural network is trained on top of it for the downstream prediction task”: is this the SVM described in the Appendix, or a neural network with one hidden layer? If it’s the former, that’s somewhat non-standard and will likely lead to results that aren’t directly comparable to the normal linear probe benchmarks

**Strengths And Weaknesses:**

Strengths:
- Augmenting diffusion models with a learned representation is a useful application that would allow more easily controllable generation
- Apply the latest in generative modeling towards improved representation learning in the semi-supervised setting is  a promising area of study
- There are a variety of great experimental ablations within the proposed approach that highlight the important components, and several additional insights (e.g. mixing uniform and random noise for faster sampling from diffusion models)

Weaknesses:
- Theoretical: Presenting this work from the perspective of score-based models makes many of the modeling decisions confusing. I think the core conditional method could be described more simply as DSM for the conditional distribution p(x|y), and then the VDRL formulation is just a VAE with a Gaussian prior (when using KL regularization), diffusion decoder p(x|y), and a learned encoder q(y|x). As you point out, learning the encoder to maximize p(x|y) leads to information-maximizing solutions, and I am thus surprised it works well! The deviation from the non-time-dependent encoder to one that is time-dependent still confuses me, and I’m not sure how to justify it theoretically? Also understanding more how the time-dependent encoder interacts with the weighting of diffusion levels could be useful? Given this time-dependent encoder it’s no longer clear that the decoder is still a diffusion model (as each score function is conditioned on a different thing). How do you sample from this time-dependent encoder diffusion model? From the experimental results, the time-dependent encoder is critical to achieve good performance. Is there a probabilistic interpretation of this time-dependent encoder? Overall, I found the presented method to be intriguing but confusing in the context of learning a generative model, and it left me wondering why the method works, and thus questioning the results.
- Experimental: while I recognize the computational demands of attempting to perform representation learning on ImageNet, accuracy numbers on CIFAR-10 are basically saturated, and the performance improvements relative to prior work on CIFAR-100 and MiniImageNet are small. There’s no error bars to indicate how significant these improvements are, and excluding a huge body of work in this area due to use of more complex augmentation schemes feels unfair. The proposed method is not permutation-invariant (it uses U-nets/convolution architectures), and thus allowing for augmentations like random crops feels fair to me. There are several methods on CIFAR-100 that outperform the proposed approach and are not discussed (because of the lack of permutation-invariant augmentations?) https://paperswithcode.com/sota/semi-supervised-image-classification-on-cifar-2

---

> ### Author Response · Authors · 2022-10-02
> **Response to Reviewer**
>
> **Probabilistic Formulation**: We agree with the reviewer’s assessment of the initial time-independent VDRL presentation. However, this provides us an opportunity to extend to time-dependent settings to obtain infinite-dimensional representation for any given sample. This proposed system is a way to augment diffusion based generative modeling for representation learning, and with time-dependent representations, it is not a generative model anymore and thus cannot be used for sampling but only for representation learning. Practically, we view the time-dependent system as a way to encode granularity-specific information in different parts of the representation trajectory, as at any time $t$, the representation $E(x, t)$ is the code that would help to denoise at the particular time, unlike $E(x)$ which is a global representation that helps to denoise at any point in the reverse process.
>
> VDRL Objective is a VAE: We don’t think that it is similar to a VAE as the encoded representation is not used for reconstruction (like in VAEs) but for denoising by conditioning the score model. We would like to ask for clarification on what the reviewer means by a decoder here, since the score-model formulation does not have a decoder.
>
> **Time-Dependent Encoder**: We do not have a more probabilistic interpretation of the time-dependent encoder system, since in general the model is not a generative model anymore as the score function gets conditioned on an encoded representation of the unperturbed input. Our interpretation of it is that it provides a more expressive encoder where the source of extra information is tailored for diffusion models that undergo a gradual denoising process.
>
> While we do not have a mathematically rigorous proof for this, the intuition behind providing the encoder with the time variable is to increase the expressiveness of the encoder in a way that is suitable for the formulation of denoising score matching where noise is gradually reduced during generation. The objective function of Equation (9) (before expectation over t) is a function of $t$. Hence, optimizing it with respect to the encoder parameters which itself is a function of $t$ allows it to decide which information from the clean data is more useful when denoising for each time step. If it was indifferent to time, it can simply ignore the time input and learn a time-independent representation. However, our experiments showed that the initial intuition holds and including the time parameter in the encoder's input actually helps.
>
> **Impact of Weights for Diffusion Levels on Time-Dependent Encoder**: We agree with the respected reviewer that an in-depth study of this interaction is enlightening but postponed it to future work due to the time and resource limitation for this work.
>
> **Layout**: We have moved the figures around so that they are closer to the text where they are mentioned in the updated revision.
>
> **Infinite-Dimensional Code**: The parameter t is theoretically continuous but in practice we choose the values it takes a priori, e.g. on a discrete regular one dimensional grid and stay with these values throughout the algorithms. A finer grained grid is closer to a continuous setting but is naturally computationally more demanding. Hence, a moderately fine grained time axis is chosen as the set of available time values. The term “infinite-dimensional” refers to the possibility of choosing as many time values as wished that is only limited by computational resources. This is in line with non-parametric models where the notion of “infinite” comes through the possibility of using unbounded number of parameters or representational information. A simple way to view it is visualizing some parameterized curve in $\mathbb{R}^2$, which is an uncountable object and yet embedded in a finite space, namely $\mathbb{R}^2$.
>
> **Constant in DSM**: We agree that the constant is the constant which is obtained in Vincent (2011). Our aim for mentioning the constant is that in traditional denoising score matching, this is an irreducible constant. However, with the additional latent-code, this constant can be reduced to 0 by learning an appropriate latent representation.
>
> **“single-layered neural network is trained on top of it for the downstream prediction task”**: We use a neural network with a single hidden layer as a standard non-linear probe on top of the representation learned.
>
> **Theoretical Justificaiton for $\sigma(t)$**: We thank the reviewer for bringing this work to our attention. We have edited the relevant sentence and revised our draft to bring this into attention.
>
> We have changed the use of the word handicapped SimCLR to SimCLR-Gauss.

---

### Decision · Action_Editors · 2022-11-01

**Recommendation:** Reject

**Comment:**

Using diffusion models and more generally generative models for representation learning presents an interesting research opportunity. This paper modifies the time-conditional score function to incorporate a learnable feature extractor for representation learning. The reviewers argue that this modification is not grounded in deep theoretical insights and does not yield strong empirical results. In addition to the discussion in the claim section above, masked image modeling (https://arxiv.org/abs/2111.06377) and denoising autoencoders are much simpler than the proposed approach, but yield very strong results given the right neural network architecture and an appropriate training recipe.

Overall, the paper lacks deep theoretical insights and does not present convincing empirical results. Unfortunately, this makes accepting the manuscript in its current form very difficult, which is also reflected in reviewers' recommendations (2x reject, 1x weak reject). The reviewers, in particular Reviewer o2f5 have some suggestions for ways to improve the paper, and I hope these suggestions can be incorporated in future revisions of this paper.

**Audience:**

The general idea of using diffusion models and more generally generative models for representation learning is interesting and likely intriguing to at least a subset of TMLR's audience. That said, per reviewers' assessment very limited theoretical insights are presented and the empirical results are not very convincing.

**Claims And Evidence:**

At least one of the claims in the paper is not grounded in clear evidence. The paper claims that:
>We evaluate the proposed approach on downstream tasks using the learned representations directly as well as using
it as a pre-training step for semi-supervised image classification, thereby improving state-of-the-art approaches for the latter.

Reviewer o2f5 questions the state-of-the-art claim as follows:

>While I recognize the computational demands of attempting to perform representation learning on ImageNet, accuracy numbers on CIFAR-10 are basically saturated, and the performance improvements relative to prior work on CIFAR-100 and MiniImageNet are small. There’s no error bars to indicate how significant these improvements are, and excluding a huge body of work in this area due to use of more complex augmentation schemes feels unfair. The proposed method is not permutation-invariant (it uses U-nets/convolution architectures), and thus allowing for augmentations like random crops feels fair to me. There are several methods on CIFAR-100 that outperform the proposed approach and are not discussed (because of the lack of permutation-invariant augmentations?)

Reviewer MPuM also indicates:

>Baselines are relatively weak and not convincing. It is known that data augmentation (e.g., cropping, color jittering) plays an important role in contrastive learning frameworks. It might not be a fair comparison to consider a simCLR framework using only Gaussian noise as data augmentation. Instead, a more proper comparison would be to compare with simCLR in a default setting and use the same data augmentation (or only Gaussian noise augmentation) for the proposed diffusion framework.

I agree with these assessments. Given that the U-net architecture used in diffusion models is typically convolutional, hence not permutation-invariant, either the method should be compared with more complex data augmentation strategies or the state-of-the-art claim be removed.